# Effect of Growth Stage on Nutrition, Fermentation Quality, and Microbial Community of Semidry Silage from Forage Soybean

**DOI:** 10.3390/plants13050739

**Published:** 2024-03-06

**Authors:** Kexin Wang, Shengnan Sun, Yilin Zou, Yongqi Gao, Zifeng Gao, Bo Wang, Yi Hua, Yalin Lu, Guofu Hu, Ligang Qin

**Affiliations:** Department of Grassland Science, College of Animal Science and Technology, Northeast Agricultural University, Harbin 150030, China; wangkexin_echo@163.com (K.W.); 19804588705@163.com (S.S.); zouyilinyl@163.com (Y.Z.); gyqqqqqqq123@163.com (Y.G.); 13030086186@163.com (Z.G.); neauwangbo1996@163.com (B.W.); qq2549188890@163.com (Y.H.); lyl1236542023@163.com (Y.L.)

**Keywords:** semidry silage, soybean, fermentation quality, nutritional quality, microbial community

## Abstract

Soybean (*Glycine max* (Linn.) Merr.) is highly suitable as animal feed. The silage quality and microbial characteristics of soybean silage are still unclear. Forage soybean (HN389), at six different growth stages (R2-R7), were used as experimental materials to investigate the changes in fermentation, nutritional quality, and microbial characteristics of semidry silage after 0, 7, 14, 30, and 45 d. As the growth period extended, the content of crude protein (CP) and crude fat (EE) gradually increased, while the neutral detergent fiber (NDF) and the acid detergent fiber (ADF) content decreased. The pH value also decreased gradually with fermentation time, accompanied by increases in the proportion of ammonia-N and the content of lactic acid (LA) and acetic acid (AA). In addition, competitive inhibition was observed in the microbial fermentation. With the process of ensiling, *Lactobacillus* became the dominant bacterial species. The results indicate that the most active stage of fermentation during ensiling occurred within the first 7 days, the fermentation and nutritional quality of the soybean forage were improved, and the optimal mowing stage was the grain stage. Comparison of the microbial abundance showed that all microorganisms entered a stable stage at 30 days of silage. After storage, the dominant bacteria were *Lactobacillus*, *Enterobacter*, and *Pantoea*.

## 1. Introduction

Soybean (*Glycine max* (Linn.) Merr.), an annual herb of the legume family, is one of the main cash crops in China [1]. With its high nutritional value, it is an important oil crop and a good protein feed crop [2]. Furthermore, soybeans are renowned for their high nitrogen content both in seeds and hay, making them superior among annual legumes in this regard [3,4]. The crude protein (CP) content in soybean hay stalks, leaves, and pods is approximately 12–14%, 19–20%, and 12–27%, respectively [5,6]. Soybean hay possesses nutritional value comparable to that of alfalfa, thanks to its abundant reserves of crude protein, crude fiber (CF), and crude fat (EE), as well as its palatability [7,8]. Research has demonstrated that soybeans have evolved from being primarily used as ruminant forage to becoming a significant feed source for piglets and a special diet for fish farming [9,10,11]. The expanding utilization of soybeans in animal feed highlights their potential value as a nutritious and adaptable resource.

Researchers have identified the significant potential of soybean silage without additives as a coarse feed [12]. The digestibility of additive-free soybean silage, soybean mixed silage, and alfalfa silage was tested with heifers. It was found that additive-free soybean silage exhibited superior digestibility compared to other roughages [13]. In recent years, research reports have highlighted the possibility of converting legume and wheat silage into semidry silage without the need for additives, thereby offering an alternative to traditional silage as a form of coarse feed [14]. However, the soybean harvest is seasonal with high accumulation, which needs a safe and efficient means of conservation for ruminants [15].

Semidry silage is a type of roughage technology that compresses harvested feed plants into packages, wrapping them in polyethylene plastic and storing them in an airless environment with a dry matter content ranging from 40% to 60% [16]. Compared with hay and traditional silage, semidry silage has been reported to offer advantages in terms of reduced dry matter and nutrient loss [17]. The buffering capacity of legume grasses poses challenges for their storage in comparison to gramineous grasses. Currently, two commonly employed methods for leguminous forage include the production of semidry silage or the addition of certain additives. However, incorporating additives escalates the actual production costs, thus emphasizing the significance of semidry silage as a pivotal approach towards advancing forage soybeans in the future. This study measured and evaluated the fermentation, nutritional quality, and microbial characteristics of semidry forage soybean silage taking into account the effect of the growth stage, which has important research significance and application value for the preparation and utilization of forage soybeans as silage.

## 2. Results

### 2.1. Nutritional Quality of Semidry Forage Soybean Silage

Table 1 shows the content of crude protein (CP), crude fat (EE), neutral detergent fiber (NDF), acid detergent fiber (ADF), and crude ash (Ash) of semidry soybean silage cut during the R3-R7 stages after 45 days of fermentation. The highest CP content (18.18%DM) was observed in the initial grain stage (R5). While the lowest Ash content (6.62%DM) was found during the initial pod stage (R3). Moreover, the lowest ADF content (2.25%DM) was recorded in the early maturity stage (R7), the lowest NDF content (38.31%DM) was observed in the grain filling stage (R6). The EE content increased gradually with the extension of the growth period, and the highest EE content (8.41%DM) was recorded in the R7 period.

### 2.2. Fermentation Quality of Semidry Forage Soybean Silage

Forage soybeans harvested during R5, R6, and R7 stages were made into semidry silage and then fermented for 45 days. The pH and ammonia-N, lactic acid (LA), acetic acid (AA), propionic acid (PA), and butyric acid (BA) contents were determined. As shown in Table 2, both LA and AA content increased gradually with increasing fermentation time. However, it was observed that during the R5 to R7 cutting period, the LA content decreased significantly at day 0 (3.57 vs. 1.96 vs. 1.77%DM), which indicated that soybean plants harvested during the later stages of growth period tend to have lower LA content in actual production, which is not favorable for silage fermentation quality. The ammonia-N content gradually increased as ensiling fermentation went on, and the proportion of NH3-N was less than 10% in all treatment groups, which met the first-class standard of semidry silage.

Considering the changes in fermentation indexes over different fermentation times, the most active stage was observed during the initial 0–7 days. During this stage, some nutrients were lost from the silage, and the production of LA and AA lowered the pH of the silage, effectively inhibiting the growth of harmful bacteria. As no silage additive was used in this experiment, a small amount of PA and BA were produced in certain treatment groups, which provided materials for the subsequent microbial population analysis in this study. These findings are valuable for the investigation of additives for soybean semidry silage and their practical application in production.

### 2.3. Microbial Population Analysis of Semidry Forage Soybean Silage

#### 2.3.1. Venn Diagram of Feature Distribution

Figure 1 presents comparisons of R2_0d, R4_0d, R6_0d, R7_0d, R2_45d, R4_45d, R6_45d, R7_45d, as well as R6_0d, 7 d, 14 d, 30 d, 45 d. The former two groups of data explained the variation in microbial abundance in semidry silage with the postponement of the cutting period. The latter group illustrated the change in microbial abundance in the soybean semidry silage as ensiling fermentation was prolonged.

The results revealed a gradual decrease in the number of features with the delay of cutting time, with the order of feature numbers for the four growth periods being R2 > R4 > R6 > R7. After 45 days of fermentation, the number of features decreased significantly in all four cutting periods compared to day 0; the total number of features across the four growth periods was reduced from 149 to 55. During the R6 period, the number of focus features exhibited a sharp decline from 0 d to 7 d, then increased, followed by a further decrease to reach the lowest level, eventually stabilizing at 45 d. The total number of features observed over the five fermentation times was 45.

#### 2.3.2. Microbial Community Analysis

Bacterial colony composition and heatmap analysis of soybean semidry silage microbial communities at different stages of cutting and fermentation are shown in Figure 2. At the phylum level (Figure 2a,d), the main dominant flora in the fermentation process of semidry silage transformed from *Cyanobacteria* and *Bacteroides* to *Firmicutes*. In addition, compared with the later stage of fermentation, the semidry silage materials at each cutting stage before fermentation contained a certain amount of *Bacteroidetes*. At the same time, the abundance of *Proteobacteria* decreased with increasing fermentation time. Compared with the experimental group during the R6 cutting period, the dominant populations of *Cyanobacteria* and *Bacteroides* basically disappeared 7 days before fermentation. The abundance of *Firmicutes* increased and the abundance of *Proteobacteria* decreased with fermentation time. At the family level (Figure 2b,e), the dominant microbial communities changed from *Oxyphotobacteria*, *Sphingomonadaceae*, and *Pseudomonadaceae* to *Lactobacillaceae* and *Enterobacteriaceae*.

In addition, at the genus level (Figure 2c,e), the dominant bacteria transformed from *Oxyphotobacteria*, *Sphingomonas*, and *Psedomonnas* to *Enterobacter* and *Lactobacilllus*. Among them, *Lactobacilllus* was dominant in the R2 and R4 periods, while the abundance in the R6 and R7 periods decreased, and *Enterobacter* replaced *Lactobacilllus* as the dominant family in the semidry silage. It can be concluded that, with the postponement of cutting time, the making of soybean semidry silage gradually becomes more difficult. The transformation of the above microbial populations was the most obvious in 0–7 d. *Pantoea* was found after 7 days of silage fermentation. *Pantoea* had antibacterial activity and could inhibit the growth of Aspergillus niger [18].

#### 2.3.3. Alpha Diversity Analysis

Figure 3 illustrates the effects of soybean mowing time and fermentation duration on the alpha diversity of the semidry silage. The results indicated that for the R2 cutting period, prior to fermentation (0 d), the Chao1, Simpson, and Shannon indexes were higher compared to the other cutting periods, with values of 440, 4.2, and 8.2, respectively. Within the R6 period, the Chao1 index was the highest for R6_0d with fermentation, reaching 250. Conversely, the Goods coverage and Simpson indexes were the lowest at 0.998 and 0.72, respectively. No significant changes were observed in Shannon’s relative change. Additionally, the Shannon index of R6_14d exhibited the highest value. There were no significant differences among the Simpson, Shannon, Goods coverage and Chaol indexes.

#### 2.3.4. Beta Diversity Analysis

The effects of soybean mowing time and fermentation time on the beta diversity of the semidry silage are shown in Figure 3. In this experiment, unweighted UniFrac PCA (Figure 4a) and PCoA (Figure 4b) analyses were conducted, revealing relatively long distance nodes for the R2, R4, R6, and R7 cutting periods. This indicates that fermentation time strongly influences the microbial community structure in soybean semidry silage. The non-metric multidimensional scaling (NMDS) analysis (Figure 4c) is an important measure of beta diversity, with the stress coefficient used to evaluate the analysis results. The results showed a stress coefficient of 0.02, indicating a significant difference in species composition between 0 d and 45 d fermentation group (*p* < 0.05).

#### 2.3.5. Advanced Analysis

Sankey plots are used to show the “flow” change of data, and the width of the branch indicates the size of the traffic. Appendix A presents the relative abundance of flora at the phylum level (middle) and the relative abundance of 26 genera (right) corresponding to different samples (left), and it also provides the species annotation information, correspondence, and proportion of the two most significant levels studied in flora diversity. Specifically, *Oxyphotobacteria* belongs to the phylum *Cyanobacteria*, and it only existed in the experimental group before fermentation (0 days). *Lactobacillus* accounted for approximately 90% of the genera belonging to the *Firmicutes* phylum, and a large number of *Lactobacillus* existed in the late fermentation (45 d) group. *Enterobacteriaceae* accounted for approximately half of the total amoebas, and a large number of *Enterobacteriaceae* existed in the experimental group during the cutting periods R6 and R7. Due to its negative impact on silage quality, the presence of *Enterobacter* led to inferior semidry silage produced in the R6 and R7 stages compared to the R2 and R4 stages.

The circos circle and bubble plot diagrams (Appendix A) were used to analyze the relative abundance differences at the genus level between the prefermentation (0 d) and postfermentation (45 d) groups. By comparing the changes in species abundance among the R6 groups (0 d, 7 d, 14 d, 30 d, and 45 d), the relative abundance of *Oxyphotobacteria_unclassified* was shown to decrease sharply 7 days before fermentation and then tended to be stable. The relative abundance of *Hymenobacter* also decreased 7 days before fermentation and disappeared completely after 14 days of fermentation. The relative abundances of *Pseudomonas*, *Sphingomonas*, and *Methylbacilli* also decreased 7 days before fermentation and then reached a stable state. The relative abundance of *Lactobacillus* increased with fermentation time and became stable after 30 days of fermentation. The relative abundance of *Enterococcus* increased with fermentation time and became stable after 14 days of fermentation. The relative abundance of *Enterobacter* increased with fermentation time and became stable after 7 days of fermentation. Based on these results, the change in the relative abundance of species during semidry silage production mainly occurred 7 days before fermentation.

To identify the species with significant differences in abundance among the different cutting periods, LEfSe was used to analyze the species abundance differences among the R2_45d, R4_45d, R6_45d, and R7_45d groups, as displayed in Appendix A. The results showed the six most abundant species in the R2_45d group, such as *Lactobacillus* sp. *TS2*, *Pediococcus acidilactici*, *Enterococcus malodoratus*, etc, and the eight most abundant species in the R4-45d group, such as *Leuconostoc Pseudomesenteroides*, etc. Uncultured *Klebsiella* sp. was the most abundant species in the R6_45d group. *Enterobacter unclassified* and *Enterobacter asburiae* were the two most abundant species in the R7_45d group. In the R2_45d group, *Lactobacillus* spp. and *Pediococcus acidilactici* inhibited the growth of undesirable microorganisms. It showed that the mowed soybeans in the R2 period had a better effect on the production of semidry silage.

To identify species with significant differences in abundance during different fermentation times, LEfSe analysis was employed on the R6_0d, R6_7d, R6_14d, R6_30d, and R6_45d groups, as shown in Appendix A. The results showed that the abundance of *Enterobacteriaceae* increased significantly after 7 days of fermentation. After 14 days of fermentation, the abundance of *Lachnospiraceae* increased significantly. After 30 days, the abundances of *Lactococcus* and *Pediococcus* became significantly elevated. The total amount of bacteria reached its peak significance level after 45 days. It can be concluded that *Enterobacteriaceae* and other harmful bacteria reached significant levels at 7 days. Subsequently, beneficial bacteria such as *Lachnospiraceae*, *Lactococcus*, and *Pediococcus* gradually reached significant levels, with their growth stabilizing after 30 days.

PICRUSt [19] established a “mapping” between flora and function. As shown in Appendix A, the presumptive biological function changes of the semidry silage microflora in the prefermentation group and the postfermentation group were verified by PICRUSt 2. All four stages (R2, R4, R6, and R7) were improved in metabolic pathways. After 45 days, the R2 group showed enhanced activity in phosphotransferase system (PTS), glycolysis/gluconeogenesis, starch and sucrose metabolism, and amino acid metabolism. In the R4 group, the metabolic activity of the amino acid metabolism, the metabolism of cofactors and vitamins, the carbohydrate metabolism, the galactose metabolism, ribosome biogenesis, the pentose phosphate pathway, and glycolysis/gluconeogenesis increased. As for the R6 group, the metabolism of cofactors and vitamins, carbohydrate metabolism, fructose and mannose metabolism, amino sugar and nucleotide sugar metabolism, pyruvate metabolism, amino acid metabolism, starch and sucrose metabolism, and galactose metabolism activity enhanced. Most metabolic pathways were improved in the R7 group, like the metabolism of terpenoids and polyketides, the biosynthesis and biodegradation of secondary metabolites, the carbohydrate metabolism, the propanoate metabolism, the nucleotide metabolism, the purine metabolism, the starch and sucrose metabolism, glycolysis/gluconeogenesis, and the amino acid metabolism.

Redundancy analysis (RDA) revealed the relationship between major microbial communities and silage quality (Appendix A). As shown in Appendix A, ph and Ammonia-N were the main factors influencing fermentation that determined the abundance structure of *Lactobacillus* in different fermentation periods. There was a strong positive correlation between *Lactobacillus*, *Sphingomonas*, and *Catiobacter*. Furthermore, the first and second axes together accounted for more than 58.66% of the total variance in the bacterial communities in terms of nutritional quality (Appendix A). Among them, NDF had a negative relationship with *Lactobacillus*; however, Ash and *Lactococcus* were positively correlated.

## 3. Discussion

### 3.1. The Effect of Mowing Period on the Nutritional Quality of Semidry Soybean Silage

In the present study, it was found that the CP content of the whole soybean plant increased gradually in the R3–R7 stages, which was consistent with the results of Miller et al. [4]. Gao et al. [20] believed that the nonprotein nitrogen (NPN) produced by plant protease degradation of proteins in the early stage of silage is unavailable nitrogen for animals, which leads to the loss of feed nutrients. The current study found that the CP content decreased slightly after 45 days of storage as semidry silage, which was consistent with the conclusions of Gao et al. [20]. Casper et al. [21] think that, for soybean, it is generally necessary to carry out semidry silage before the R6 stage. When the soybean is fully mature, the digestibility of forage grass will be reduced. The decrease in CP content in the R6 and R7 groups in this study was consistent with the conclusions of Casper et al. [21]. The content of ammonia nitrogen and the relative abundance of microorganisms in the R6 and R7 groups increased significantly after fermentation, the quality of the semidry silage without additives in the R6 and R7 groups was relatively poor. ADF is a measure of the indigestible components of fiber [22]. This study showed that with the postponement of the growth period, the NDF content did not change significantly, but the ADF content decreased. Some microbes consume WSC and CP during silage fermentation to produce cellulase, which collapses NDF and ADF [23]. As digestibility is negatively correlated with cell wall lignin and fiber contents, reducing fiber content is helpful to developing feed crops with higher digestibility [24]. This study showed that soybean semidry silage has better feed digestibility than hay, which is similar to the results of Vurarak [25]. EE is not only one of the important nutritional indexes of soybean but also the main substance of forage energy reserves. In this experiment, the EE content of semidry soybean silage increased gradually from R3 to R7, and there was no significant change in EE content in the five cutting periods.

### 3.2. The Effect of Mowing and Fermentation Stages on Fermentation Quality of Semidry Soybean Silage

The pH value of the three groups of R5–R7 materials decreased gradually with fermentation time in the process of semidry silage production. Although the final pH value of all three groups was above 4.5, pH is not an indicator of the quality of semidry silage, and the quality of semidry silage can be maintained at higher pH values [26]. The ratio of ammonia-N reflects the degree of decomposition of proteins and amino acids. Ammonia-N is a crucial index to assess the fermentation quality of silage [27]. The higher the value, the greater the degree of decomposition, and the worse the quality of the silage [28]. In this experiment, the proportion of ammonia-N in the process of semidry silage fermentation of R5–R7 materials increased with fermentation time, and the ratio of ammonia-N was approximately 5% after 45 days. LA is the desired fermentation product in silage, produced primarily by homolactic fermentation (*Lactobacillus*) [29]. The higher the LA content, the better the fermentation quality of the silage. The reason for this correlation is that *Lactobacillus* does not have proteolytic enzymes and, thus, does not decompose protein into raw materials, so there is little protein loss in high-quality silage. With the increase in fermentation days, the content of LA in the dry matter increased gradually. With the postponement of cutting time, the content of *Lactobacillus* attached to the plant surface decreased significantly. This showed that soybean with late growth periods is relatively unsuitable for silage in actual production, and the addition of some silage additives containing *Lactobacillus* may be beneficial for the quality of the silage [15]. In this study, the change trend of AA content was the same as that of LA content, and AA has strong antifungal properties [29]. Silage fermentation is a dynamic fermentation process that involves the joint participation of some beneficial bacteria, such as *Lactobacillus*, and some microorganisms, such as fungi, saccharomycetes, *Clostridium butyricum*, and other microorganisms that are not conducive to silage fermentation. Among those organisms, the fermentation conducted by *Propionibacterium*, *Clostridium butyricum*, and *Clostridium* can produce volatile acids, such as propionic acid and butyric acid, which are not conducive to silage quality [30].

### 3.3. The Effect of the Mowing and Fermentation Stages on the Microbial Structure of Semidry Soybean Silage

Polley et al. [31] believed that the higher the abundance of the dominant flora, the more homogenized the flora. Through the analysis of alpha and beta diversity, the abundance and diversity of bacteria peaked at 30 days. A large number of microorganisms declined and died after 30 days. This also indicated that the microbial community was more stable [32].

Before storage, the dominant bacteria were *Oxyphotobacteria*, *Sphingomonas*, and *Pseudomonas*. However, McGarvey et al. [33] found that the main bacteria before alfalfa silage were *Erwinia*, *Escherichia*, *Pseudomonas*, and *Pantoea*. The results showed that there were some differences in microbial diversity among different silage materials. In the process of silage fermentation, *Lactobacillus* propagated most rapidly and gradually became the dominant bacteria. The key to the success of silage fermentation is that lactic acid bacteria quickly acquire the dominant position by consuming the nutrients in the silage and establishing a low pH environment to limit the protease and microorganism bacterial community [29]. In addition, *Lactobacillus* and *Pediococcus* promoted each other in the process of semidry silage fermentation, which was similar to the results of Ogunade et al. [34]. After storage, the dominant bacteria were *Lactobacillus*, *Enterobacter*, and *Pantoea*. *Enterobacteriaceae* and other bad bacteria reached significant levels at 7 days. Then, beneficial bacteria, such as *Lachnospira*, *Lactococcus*, and *Pediococcus*, gradually reached significant levels, and their mass reproduction tended to be stable after 30 days.

At the gene level, the changes in metabolic pathways in different stages were analyzed based on the EC database. The alanine metabolic pathway, fructose metabolism, and glycolysis were active in the later stage of silage. Furthermore, inhibition among microorganisms was observed at this stage. In addition, by comparing the flora structure before and after silage, we also observed the decline and death of some low-abundance pathogenic bacteria. For example, *Pseudomonas* can cause bacteremia and urinary tract infections in animals or humans [35]; *Klebsiella* can cause pneumonia, hysteritis, mastitis, and other suppurative inflammation in animals [36]; and *Flavobacterium* can cause pneumonia, meningitis, septicemia, and other diseases [37]. The number of these bacteria decreased significantly after silage.

## 4. Materials and Methods

### 4.1. Raw Materials and Silage Preparation

The tested material was a forage soybean strain (HN389), which was collected in Acheng District (45°52′ N, 127° 05′ E), in Harbin, Heilongjiang Province. The experiments were carried out on 10 June 2020, in the experimental plot of Northeast Agricultural University in Harbin, Heilongjiang Province (45°75′ N,126°73′ E). A randomized group design was used, with protected rows planted around the groups. Each district was divided into 3 plots (repetition), each with an area of 20 m × 10 m, a ridge width of 65 cm and a planting density of 40 plants per square meter. A precision dibbler was used for dibbling, and the sowing depth was 1.5~2 cm after suppression. Basic fertilizer 22 g/m^2^ was applied: pure potassium sulfate compound fertilizer 12-18-13 (total nutrients > 45%, Wuxi Baoli Chemical Fertilizer Co., Ltd, Wuxi, China).

All soybean samples in the 1 m × 2 m plot were cut at the full flowering stage (R2), initial pod stage (R3), full pod stage (R4), initial grain stage (R5), grain filling stage (R6), and early maturity stage (R7), and the stubble height was 5 cm.

The harvested plants were dried until the moisture content reached 45~55%, cut short (approximately 2 cm), and vacuum sealed in polyethylene silage bags of 500 g each. The samples were stored at room temperature away from light. The fermentation index experiments identified three cutting periods with higher yields, the R5, R6, and R7 groups. In the nutritional index test, the R3 and R4 groups were combined and used as the control group. Microbial population analysis identified four experimental groups with a long difference in cutting period: the R2 group, R4 group, R6 group, and R7 group. After 0 d, 7 d, 14 d, 30 d, and 45 d, the nutritional quality, fermentation quality, and microbial population were analyzed.

### 4.2. Nutrition and Fermentation Indexes

After 0 d, 7 d, 14 d, 30 d, and 45 d of semidry silage fermentation, the bagged silage was placed in an oven for drying (15 min, 105 °C) and then dried to a constant weight (24 h, 65 °C). Finally, the silage was ground into grass powder in silage bags for the determination of subsequent nutritional indexes (40 mesh). The crude protein (CP) content was analyzed using a Kjeltec 8400 nitrogen analyzer (FOSS Analytical AB, Hoganas, Sweden) according to the methods of the Association of Official Analytical Chemists [38]. The neutral detergent fiber (NDF) and acid detergent fiber (ADF) contents were determined according to according to the method of Van Soest et al. [39] (ANKOM2000 fiber analyzer, ANKOM Technology, Macedon, NY, USA). The crude fat (EE) content was determined by an ANKOM automatic oil quantitative analysis and extraction device (ANKOMEXTRACTOR XT15l, ANKOM Technology, Macedon, NY, USA). The crude ash (Ash) content was determined by the high-temperature burning method in the muffle furnace.

A 20 g sample from each bag of semidry soybean silage was added to 180 mL distilled water and stirred for 1 min. The sample extract was obtained by filtering with coarse gauze and filter paper. The pH value of the filtrate was determined by a pH meter. The other part was centrifuged (4500× *g*, 15 min, 4 °C), and the supernatant was retained for organic acid and ammonia-N analysis. Ammonia-N was determined by phenol-sodium hypochlorite colorimetry [40]. The concentrations of lactic acid, acetic acid, propionic acid, and butyric acid were determined by high-performance liquid chromatography (HPLC) (Waters600, Waters Technology (Shanghai) Co., Ltd., Shanghai, China).

### 4.3. Microbial Index Analysis

Total DNA from various samples was extracted using a DNA Kit (D4015, Omega, Inc., Waltham, MA, USA) according to the manufacturer’s instructions. The total DNA was eluted in 50 μL of elution buffer and stored at −80 °C. The extraction quality of the DNA was detected by agarose gel electrophoresis, and the DNA was quantified by ultraviolet spectrophotometry. The PCR amplification was performed in a mixture containing 25 μL of total volume, which included 12.5 μL of PCR premix, 50 ng template DNA, 2.5 μL of each primer, and PCR-level water regulation. The 16S rDNA V3-V4 regions were amplified using primers 341F (CCTACGGGNGGCWGCAG) and 805R (GACTACHVGGGTATCTAATCC). Ultrapure water was used throughout the whole process of DNA extraction. PCR products were quantified by AMPureXTbeads (Beckman Coulter Genomics, Danvers, MA, USA) and Qubit (Invitrogen, Waltham, MA, USA). Amplicon pools were used for sequencing, and the size and number of amplicon libraries were evaluated using library quantification kits from Agilent2100 Bioanalyzer (Agilent, Santa Clara, CA, USA) and Illumina (Kapa Biosciences, Woburn, MA, USA), respectively. The libraries were sorted on a NovaSeq PE250 platform.

The samples were sequenced on an Illumina NovaSeq platform provided by LC-Bio. Under specific filtering conditions, the quality of the original read data was filtered according to fqtrim (v0.94) to obtain high-quality clean tags. Research software was used to filter chimeric sequences (v2.3.4). Using DADA2 to demodulate, the feature table and feature sequence were obtained. Diversity was calculated by normalizing 7 to the same random sequence. Then, according to the SILVA (Release132) classifier, the feature abundance was normalized using the relative abundance of each sample.

### 4.4. Statistical Analyses

The nutrition index and fermentation quality data were statistically analyzed and plotted using WPS2021 (v11.1.0.11045) and SPSS21 (v21.0) software. All indicators of microorganisms were analyzed by QIIME2. Alpha diversity was determined using WPS2021 statistics and mapping. Beta diversity was calculated by QIIME2 and plotted by R package (v3.5.2). Using SILVA (Release 132, https://www.arb-silva.de/documentation/release-132/, accessed on 5 July 2020) and the NT-16S database for species classification and subsequent analysis. The PICRUSt analysis and redundancy analysis (RDA) were performed on a free online platform omicstudio (https://www.omicstudio.cn/, assessed on 13 April 2020). Other diagrams were also implemented using the R package (v3.5.2).

## 5. Conclusions

This study showed that different cutting periods and fermentation stages affect the content of nutrients in silage. The fiber content decreased in this experiment, which indicated that the digestion and absorption rates of soybean silage were higher than that of soybean hay. In actual production, soybean materials with late growth periods are relatively unsuitable for silage production, and the addition of some silage additives containing *Lactobacillus* may be beneficial for silage quality. The differences and dynamic changes in microbial population structure were also discussed in this study. *Lactobacillus* became the most dominant bacteria after semidry storage.

## Figures and Tables

**Figure 1 plants-13-00739-f001:**
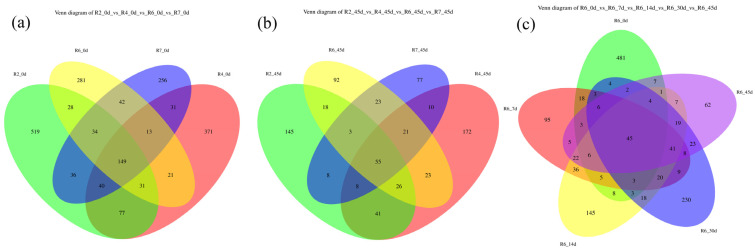
Venn diagram of feature distributions. Note: Different colors represent different mowing periods and silage times. Data of all groups are described as the average (n = 3), the following figures are the same.

**Figure 2 plants-13-00739-f002:**
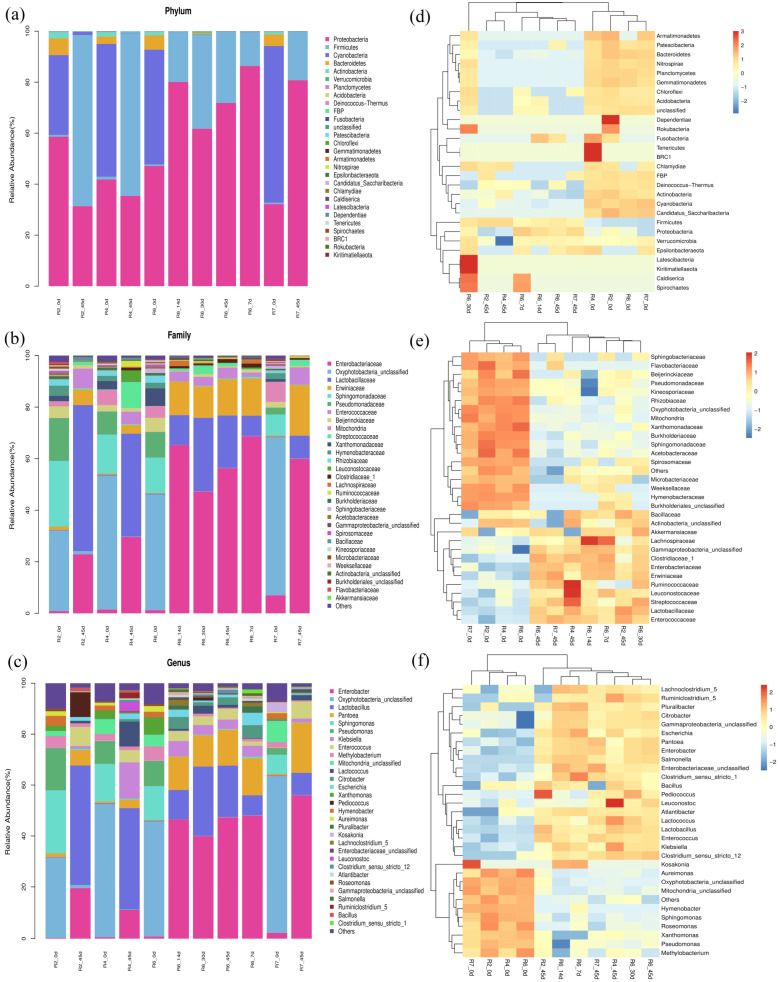
Taxonomic analysis and heatmap analysis of soybean haylage flora at different cutting stages and different fermentation stages. (**a**–**c**) Taxonomic analysis at the phylum, family, and genus level during semidry ensiling, (**d**–**f**) heatmap analysis at the phylum, family, and genus level during semidry ensiling.

**Figure 3 plants-13-00739-f003:**
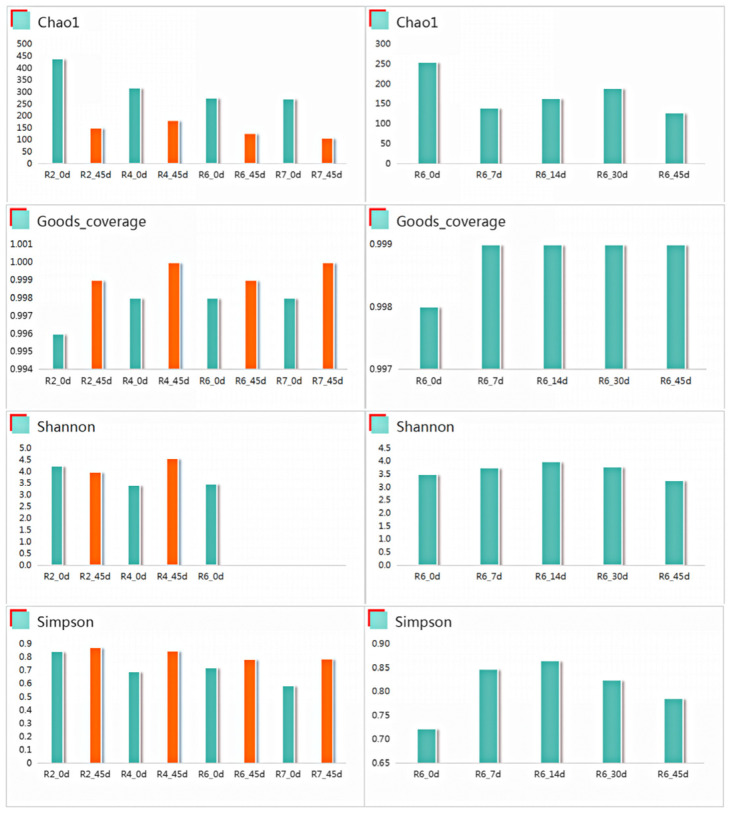
Effects of soybean cutting period and fermentation time on alpha deversity of semidry silage.

**Figure 4 plants-13-00739-f004:**
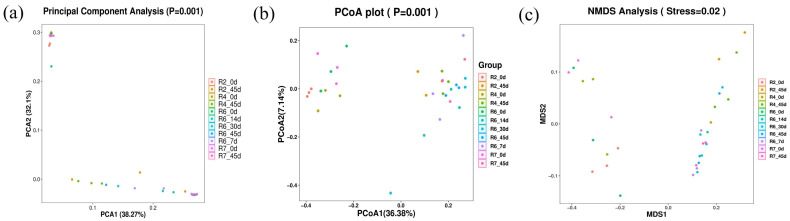
Effects of mowing period and fermentation time on beta diversity of soybean semidry silage.

**Table 1 plants-13-00739-t001:** Nutritional quality of semidry silage made from forage soybean at different reproductive periods (%DM).

Project	CP	NDF	ADF	EE	Ash
R3	17.06 ^bc^	41.36 ^ab^	27.23 ^bc^	4.14 ^f^	6.62 ^bcd^
R4	17.15 ^bc^	42.48 ^ab^	25.32 ^cd^	5.21 ^def^	7.92 ^a^
R5	18.18 ^b^	41.01 ^ab^	24.16 ^cd^	6.17 ^de^	8.13 ^a^
R6	16.32 ^bc^	38.31 ^b^	23.75 ^cd^	6.68 ^cd^	7.04 ^bc^
R7	17.83 ^b^	39.62 ^ab^	22.25 ^d^	8.41 ^b^	7.01 ^bc^

Note: R3: initial pod stage, R4: full pod stage; R5: initial grain stage; R6: grain filling stage; R7: early maturity stage. CP: crude protein; NDF: neutral detergent fiber; ADF: acid detergent fiber; EE: crude fat; Ash: crude ash; DM: dry matter. The same row (a–f) represents the significant difference in samples under the same treatments. Values with different letters in the same rows are significantly different (*p* < 0.05).

**Table 2 plants-13-00739-t002:** Fermentation quality of HN389 converted into semidry silage at different growth periods (% DM, except pH).

Handle	pH	Ammonium-N (%TN)	Lactic Acid	Acetic Acid	Propanoic Acid	Butanoic Acid
R5-0 d	6.23 ^a^	0.10 ^h^	3.57 ^abcd^	1.51 ^h^	0.26 ^ab^	0.00 ^e^
R5-7 d	5.25 ^ab^	2.99 ^g^	5.37 ^a^	2.35 ^fg^	0.11 ^b^	0.00 ^e^
R5-14 d	5.22 ^ab^	3.76 ^cd^	4.84 ^ab^	3.04 ^cdef^	0.23 ^ab^	0.20 ^bcde^
R5-30 d	5.05 ^abc^	4.28 ^a^	4.37 ^abcd^	3.78 ^abc^	0.29 ^ab^	0.46 ^bcde^
R5-45 d	4.97 ^abc^	5.25 ^a^	5.16 ^ab^	4.48 ^a^	0.28 ^ab^	0.68 ^abcd^
R6-0 d	6.20 ^a^	0.23 ^h^	1.96 ^cd^	1.72 ^gh^	0.09 ^b^	0.00 ^e^
R6-7 d	5.10 ^abc^	2.88 ^g^	2.91 ^abcd^	2.54 ^ef^	0.17 ^b^	0.09 ^cde^
R6-14 d	5.05 ^abc^	4.27 ^ef^	4.31 ^abcd^	2.96 ^def^	0.25 ^ab^	0.13 ^cde^
R6-30 d	5.07 ^abc^	6.76 ^bc^	3.83 ^abcd^	3.70 ^bcd^	0.38 ^ab^	0.81 ^ab^
R6-45 d	4.81 ^abc^	7.36 ^b^	4.64 ^abc^	4.01 ^ab^	0.30 ^ab^	1.20 ^a^
R7-0 d	6.27 ^a^	0.18 ^h^	1.77 ^d^	1.51 ^h^	0.06 ^b^	0.00 ^e^
R7-7 d	5.20 ^ab^	2.29 ^g^	2.59 ^abcd^	2.57 ^ef^	0.13 ^b^	0.05 ^de^
R7-14 d	5.01 ^abc^	3.40 ^fg^	2.48 ^bcd^	2.73 ^ef^	0.37 ^ab^	0.05 ^de^
R7-30 d	4.72 ^bc^	3.42 ^fg^	3.48 ^abcd^	3.26 ^cde^	0.27 ^ab^	0.31 ^bcde^
R7-45 d	4.57 ^c^	4.66 ^de^	3.50 ^abcd^	3.69 ^bcd^	0.60 ^a^	0.69 ^abc^

Note: TN: total nitrogen; 0 d: no silage control; 7 d: samples opened at 7 d; 14 d: samples opened at 14 d; 30 d: samples opened at 30 d; 45 d: samples opened at 45 d. The same row (a–h) represents the significant difference in samples under the same treatments. Values with different letters in the same rows are significantly different (*p* < 0.05).

## Data Availability

Date are stored in the National Microbiology Data Center (NMDC). The URL is https://nmdc.cn/ (accessed on 11 January 2024). The periodical attachment number: NMDCX0000263.

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
