# Peer review of "Effect of Growth Stage on Nutrition, Fermentation Quality, and Microbial Community of Semidry Silage from Forage Soybean"

_plants, 2024, doi:10.3390/plants13050739_

Round 1

Reviewer 1 Report

Comments and Suggestions for Authors

The manuscript " Effect of growth stage on nutrition, fermentation quality and  microbial community of semidry silage of forage soybean" is interesting for the scientific community but some aspects must be improved before considered for publication.

1. The purpose of the work should be rewritten.

2. Table 2 shows a great deal of data. It is unreadable for the reader.

3. All graphs are illegible. They should be improved.

4. The Materials and Methods chapter lacks information on the analysis performed with PICRUST 2.

4. The analysis of microbial communities should first include a taxonomic analysis and then a biodiversity analysis. This should be revised in the manuscript.

5. Section 2.3.4 Species analysis refers to other taxonomic levels (genus, phylum etc.) The title is wrong.

6. Data on the quality of sequencing and bioinformatics analysis (number of sequences, chimeras etc.) are missing.

7. Line 169-170  This is just an assumption not supported by your own research

Author Response

Response to the comments from editor:

The manuscript " Effect of growth stage on nutrition, fermentation quality and  microbial community of semidry silage of forage soybean" is interesting for the scientific community but some aspects must be improved before considered for publication.

Q1. The purpose of the work should be rewritten.

A1: Thank you very much for your suggestion, which has been revised in lines 54-64 of the manuscript, and the revised content is as follows: “The buffering capacity of legume grasses poses challenges for their storage in comparison to gramineous grasses. Currently, two commonly employed methods for leguminous forage include the production of semidry silage or the addition of certain additives. However, incorporating additives escalates the actual production costs, thus emphasizing the significance of semidry silage as a pivotal approach towards advancing forage soybeans in the future. This study measured and evaluated the fermentation, nutritional quality, and microbial characteristics of semidry forage soybean silage about the effect of the growth stage, which has important research significance and application value for the silage preparation and utilization of forage soybeans.” The above content has been highlighted in the text.

Q2. Table 2 shows a great deal of data. It is unreadable for the reader.

A2: Thank you very much for your suggestion, Table 2 has been simplified and modified in the manuscript, and Table 1 has also been modified to ensure the consistency of the whole paper.

Q3. All graphs are illegible. They should be improved.

A3: I'm sorry for the problem. There may be problems with images being compressed and not clear in word documents. I have added an image compression package to ensure that the images you see are clear enough.

Q4. The Materials and Methods chapter lacks information on the analysis performed with PICRUST 2.

A4: Thank you very much for your suggestions, which have been supplemented and marked in the "4.4. Statistical analyses" part of the manuscript. In addition, the listed website will provide a summary of the pathways corresponding to EC to help the experimenter find the pathways corresponding to EC.

Q5. The analysis of microbial communities should first include a taxonomic analysis and then a biodiversity analysis. This should be revised in the manuscript.

A5: Thanks for your reminding, I have adjusted the positions of "2.3.2" , "2.3.3" and "2.3.4" in the article to ensure that taxonomic analysis is shown first, followed by biodiversity analysis.

Q6. Section 2.3.4 Species analysis refers to other taxonomic levels (genus, phylum etc.) The title is wrong.

A6:Sorry for this problem, the title has been revised to "Microbial community analysis" and marked in red on line 130 of the manuscript.

Q7. Data on the quality of sequencing and bioinformatics analysis (number of sequences, chimeras etc.) are missing.

A7: I'm sorry, I have uploaded the original Data. The specific website address and login number are available in the "Data Availability Statement" section of the manuscript.

Q8. Line 169-170  This is just an assumption not supported by your own research

A8: Thank you for your question. What we want to express is that in the 0-7 d of semidry silage, the change and fluctuation of microbial community were the largest, and the disappearance and uniform growth of microorganisms were the most obvious. After that, the change and fluctuation of microbial community were small, and the overall state tends to be stable. Our use of the word "active" may have misled you, and we have now changed the word to "obvious".

Reviewer 2 Report

Comments and Suggestions for Authors

 The manuscript evaluates the quality and microbial characteristics of soybean silage at six different growth stages after 0, 7, 14, 30, and 45d of ensiling. The manuscript is within the scope of the manuscript. I have some minor comments before my final decision. 

Authors should focus on the novelty of the topic.

More recent articles should be cited.

In Table 1, the authors should include the nutrient contents of R2.

Statistical analysis should be detailed. 

More detail could find in attachment.

Author Response

Response to the comments from editor:

The manuscript evaluates the quality and microbial characteristics of soybean silage at six different growth stages after 0, 7, 14, 30, and 45d of ensiling. The manuscript is within the scope of the manuscript. I have some minor comments before my final decision.

Q1. What is the main question addressed by the research?

The manuscript answers an important question about the effect of the growth stage of soybean on the fermentation, nutritional quality, and microbial characteristics after 0, 7, 14, 30, and 45d.

A1: Soybean silage has a high protein content, but the buffering capacity of leguminous grasses makes it more difficult to store compared to gramineous grasses. The manuscript measured and evaluated the effect of the growth stage of soybean on the fermentation, nutritional quality, and microbial characteristics after 0, 7, 14, 30, and 45d. which has important research significance and application value for the silage preparation and utilization of forage soybeans.

Q2. What parts do you consider original or relevant for the field? What specific gap in the field does the paper address?

The original part of the manuscript evaluates soybean forage as a feed, with a focus on soybean forage silage. The main gap that this paper addresses is evaluating unconventional feeds to overcome the problem of feed shortage.

A2: In the original part, semi-dry silage soybeans were harvested in different cutting periods to find the best cutting period of silage quality. The main problem solved in this paper is to treat unconventional feed (feed soybean) by means of semidry silage in order to find the best cutting period, solve the problem of feed shortage, and control the actual cost without adding exogenous additives.

Q3. What does it add to the subject area compared with other published material?

Not more; however, evaluating the microbial characteristics of the silage is a very important issue.

A3: In this experiment, soybean was treated by semidry silage without adding exogenous additives in order to control the actual production cost, and the silage quality and microbial community changes of forage soybean in different growth periods were investigated.

Q4. What specific improvements should the authors consider regarding the methodology? What further controls should be considered?

Some recommendations have been included in my previous evaluation.

A4: Thank you very much for your suggestions, the "Statistical analysis" section has been revised in the manuscript to ensure its integrity, and it has been marked. In this experiment, we only explore the differences of semi-dry silage soybean in different growth periods, and no other additives are added during cultivation and treatment. In the future, we will adjust water and fertilizer management during soybean cultivation or use additives in silage according to the results of this experiment.

Q5. Please describe how the conclusions are or are not consistent with the evidence and arguments presented.

Please also indicate if all main questions posed were addressed and by which specific experiments. The conclusions seem good and are parallel with the obtained results. Moreover, all the main questions posed were addressed.

A5: The conclusions of this experiment are consistent with the evidence and arguments presented. The results of nutritional and fermentation indexes can guide the timely mowing operation in actual production, and the microbial community analysis can be carried out to link the change of silage quality with the change of microbial community, so as to provide support for the change of silage quality. All the main questions posed were addressed.

Q6. Are the references appropriate?

Yes; however, the list needs to be updated with more recent references. I already mentioned this in my revision.

A6: Thank you very much for your suggestion. Thank you very much for your advice. I have added a number of recent articles to the references and marked revisions in the manuscript.

Q7. Please include any additional comments on the tables and figures and the quality of the data.

No more than those previously mentioned.

A7: The table in the manuscript has been modified to ensure the convenience of reading. The picture is not clear enough due to the compression phenomenon in the document, and we have uploaded the compression package to ensure the clarity of the picture

Q8. Authors should focus on the novelty of the topic.

A8: Thank you very much for your advice. In this paper, we started from the silage quality of unusual diet, and analyzed the silage quality and microbial community of semi-dried silage soybean, and got the changes in different mowing periods, in order to alleviate the problem of feed shortage by feeding soybean.

Q9. In Table 1, the authors should include the nutrient contents of R2.

A9: We are very sorry that we did not test the nutritional indexes in R2 (full flowering stage) due to the average growth of soybeans. R3 and R4 periods were taken as the control group and five periods with high yield were selected for the experiment.

Round 2

Reviewer 1 Report

Comments and Suggestions for Authors

Unfortunately, figures 5-7 are still illegible to the reader. I suggest putting them in better quality in supplementary materials.

Author Response

Q: Unfortunately, figures 5-7 are still illegible to the reader. I suggest putting them in better quality in supplementary materials.

A: Thank you very much for your advice. Because there are too many words in figures 5-7, it may be difficult to see clearly. I have put these three figures in the supplementary materials to ensure the clarity of the figures. And modify the corresponding position in the manuscript.